# Diagnostic performance of serum metabolites biomarker associated with colorectal adenoma: a systematic review

Maryam Fatimah Abu Bakar[1], Siok Fong Chin[2], Suzana Makpol[3], Jen Kit Tan[3] and Azmawati Mohammed Nawi[1]

[1] Department of Public Health Medicine, Faculty of Medicine, Universiti Kebangsaan Malaysia, Cheras, Kuala Lumpur, Malaysia

[2] UKM Medical Molecular Biology Institute (UMBI), UKM Medical Centre, Cheras, Kuala Lumpur, Malaysia

[3] Department of Biochemistry, Faculty of Medicine, Universiti Kebangsaan Malaysia, Cheras, Kuala Lumpur, Malaysia

Corresponding author
Azmawati Mohammed Nawi, azmawati@ppukm.ukm.edu.my

## ABSTRACT

Evidence on serum biomarkers as a non-invasive tool to detect colorectal adenoma (CRA) in the general population is quite promising. However, the sensitivity and specificity of these serum biomarkers in detecting disease are still questionable. This study aimed to systematically review the evidence on the diagnostic performance of serum biomarkers associated with CRA. Database searches on PubMed, Scopus, and WoS from January 2014 to December 2023 using PRISMA guidelines resulted in 4,380 citations, nine of which met inclusion criteria. The quality of these studies was assessed using the QUADOMICS tool. These studies reported on 77 individual/panel biomarkers which were further analysed to find associated altered pathways using MetaboAnlyst 5.0. Diagnostic accuracy analysis of these biomarkers was conducted by constructing a receiver operating characteristic (ROC) curve using their reported sensitivity and specificity. This review identified six potential serum metabolite biomarkers with $0.7 < AUC < 1$. Benzoic acid, acetate, and lactate significantly differentiate CRA *vs.* normal, while adenosine, pentothenate, and linoleic acid are highly remarkable for CRA *vs.* CRC. The five most affected pathways for CRA *vs.* normal are glycoxylate and dicarboxylate metabolism; alanine, aspartate, and glutamate metabolism; aminoacyl-tRNA biosynthesis; D-glutamine and D-glutamate metabolism; and nitrogen metabolism. Meanwhile, pyruvate metabolism, glycolysis/gluconeogenesis, glycerolipid metabolism, citrate/TCA cycle, and alanine, aspartate, and glutamate metabolism were found to be altered in CRA *vs.* CRC. However, the association of suggested serum metabolites and altered pathways is still unknown. Despite promising emerging evidence, further validation studies in a diverse population with standardized methodology are needed to validate the findings.

## INTRODUCTION

Colorectal adenoma (CRA) also known as colorectal polyps is an abnormal cell proliferation found in the patient's colon/rectal during a colonoscopy procedure. It is categorized as a benign cell with the potential to develop into a cancerous stage (*Strum, 2016*). For benign

CRA, polypectomy in the early stage of this disease is efficient for long-term prevention (*Liu et al., 2023*). Based on longitudinal analysis of CRA in patients by computerized tomography colonoscopy (CTC) procedure, it is found that growing adenomas are prone to develop into high-risk adenomas and then become colorectal cancer (*Pickhardt et al., 2018*; *Liu et al., 2023*).

CRC ranked as the second cancer with high mortality and incidence rates reported around the world (*Sung et al., 2021*). According to *Li et al. (2020)*, the risk of incident CRC increased in individuals with large serrated polyps (SPs) by 30.2% in 3 years or more after the colonoscopy. It begins with the malignant transformation of benign adenomas. When adenomas are large or multiple, the risk of subsequent cancer is highest (*Strum, 2016*). A major challenge for reducing CRC incidence and mortality rates is to detect patients carrying high-risk premalignant adenomas with minimally invasive testing (*Ivancic et al., 2019*).

Previous studies reported significant changes in the level of several metabolites in patients carrying premalignant colonic adenomas (*Ivancic et al., 2019*). Several efforts to discover serum metabolite biomarkers also have been made and a few potential serum metabolite biomarkers for CRC have been reported. However, serum metabolite biomarkers associated with colonic adenomas and their diagnostic performance are rarely studied (*Liu et al., 2023*). Hence, this review conducted a systematic search to identify the diagnostic accuracy of serum metabolites associated with CRA.

## MATERIALS & METHODS

This systematic review is prepared following the PRISMA (Preferred Reporting Items for Systematic Reviews and Meta-Analyses) updated guidelines (*Page et al., 2021*). A complete search protocol was registered into PROSPERO [ID: CRD42023457807]. Part of the protocol was amended as follows: 1. The study duration was changed from January 2014 to December 2023. 2. For each serum biomarker, the pooled AUC was not calculated due to the non-overlapping of serum metabolites reported between the studies.

### Search strategy

Literature searches were performed in three electronic databases: PubMed, Web of Science (WoS), and Scopus to obtain a comprehensive finding of unique citations of related articles. Three databases were selected to increase the validity and quality of the results and to minimize selection bias as suggested by *Vassar et al. (2017)*. The search strategy was designed by three reviewers (MFAB, AMN, and SM) and conducted by MFAB, AMN, CSF, TJK, and SM. Searching keywords including: "metabolites" OR "metabolism product" OR "metabolic product" OR "metabolomics" or "metabolome" AND "marker" OR "biomarker" OR "biological marker" OR "biological signature" AND "blood" OR "serum" OR "circulating" AND "human" AND "Sensitivity" AND "Specificity" AND "diagnosis" OR "screening" OR "testing" OR "detecting" AND "Colorectal adenoma" OR "Colonic Polyps" OR "Polyps". Two reviewers (MFAB and AMN) independently assessed titles and abstracts of all abstracts as part of the primary screen. A secondary screen of titles

and abstracts was then conducted by a further two reviewers (SM and CSF). The results were analyzed by MFAB, SM, and AMN and confirmed by TJK and CSF.

### Eligibility criteria

Studies published between 2014 and 2023, were included to ensure that all newly published evidence on diagnostics accuracy of potential serum biomarkers for colorectal adenoma screening. The review was limited to studies on serum samples from humans, published in English, and addressed the finding of serum metabolite biomarkers in detecting CRA with sensitivity and specificity.

### Exclusion criteria

Review articles, conference abstracts, studies without a complete set of data, and articles that did not mention CRA, or serum metabolites in the title or abstract were excluded. In addition, the study was limited to serum biomarkers as this type of sample is easily obtained, hence, all other sources of CRA metabolites biomarkers such as stool or urine, were excluded. Studies with no definition of the role of serum biomarkers in colorectal adenoma diagnosis were also excluded.

### Quality assessment

The quality of each publication was evaluated by two independent reviewers (MFAB and AMN) and confirmed by the other two authors (SZ and CSF). QUADOMICS was used to assess the methodological quality of the selected studies. The quality of the studies was summarized by the percentage of applied criteria scored positively (Table S1).

### Data extraction

All articles were screened by two authors (MFAB, and AMN) and any disagreement was reached by consensus or involvement of CSF and SM. Data were extracted by two authors (MFAB, and AMN) followed by validation by SM, TJK, and CSF. The articles/studies were selected based on inclusion and exclusion criteria.

The following information was extracted from all included studies: First named author, year of publication, participant country, sample type used, number of participants, suggested serum metabolites associated with CRA, sensitivity, specificity, and type of analysis instrument as suggested by *Contreras et al. (2023)*.

### Data synthesis

Identified serum metabolite biomarkers for CRA were extracted from included studies. Enrichment pathway analysis was conducted using MetaboAnalyst 5.0 for all individual/panels of metabolites reported for CRA. The sensitivity and specificity of the serum biomarkers from the included studies were extracted and logistically transformed, and then a linear regression line was fitted through the data points. This line was back-transformed to obtain the ROC curve, which is a compact description of the accuracy of the diagnostic test in many populations.

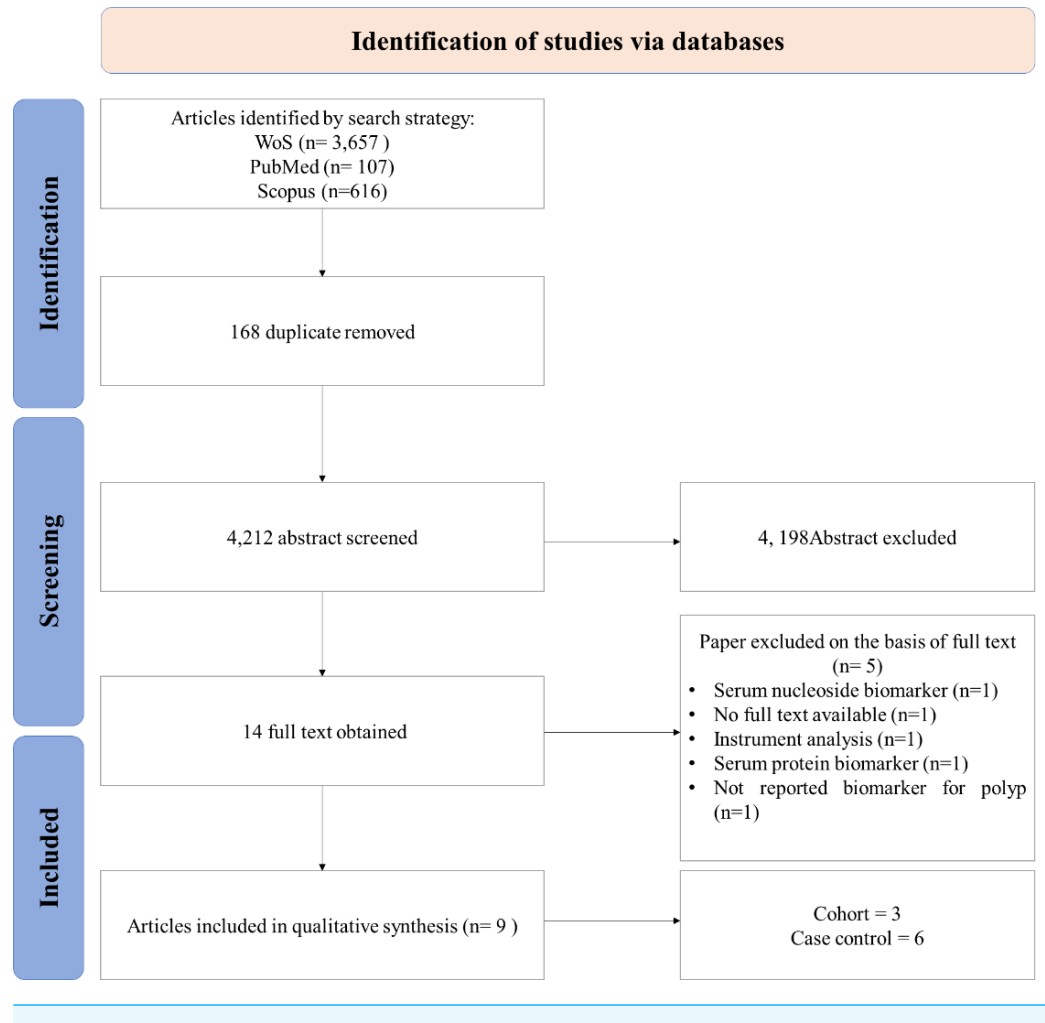

**Figure 1** **PRISMA flow diagram.**

## RESULTS

### Study selection

A PRISMA diagram of studies selected for this systematic review is summarized in Fig. 1. The search strategy identified 4,212 suitable abstracts, from which 4,198 were excluded by review of the title and abstract during the primary and secondary screens, as they did not meet the eligibility criteria. Full-text articles were obtained for 14 studies. A total of nine articles (*Guo et al., 2023*; *Tevini et al., 2022*; *Gu et al., 2019*; *Liu et al., 2023*; *Chen et al., 2017*; *Long et al., 2017*; *Uchiyama et al., 2017*; *Farshidfar et al., 2016*; *Zhu et al., 2014*), examining potential serum metabolites biomarker for CRA were included in this review for data extraction and analysis (Table 1).

Three of the studies (*Tevini et al., 2022*; *Liu et al., 2023*; *Chen et al., 2017*) used the cohort study design while the remaining six studies (*Guo et al., 2023*; *Gu et al., 2019*; *Long et al., 2017*; *Uchiyama et al., 2017*; *Farshidfar et al., 2016*) used a case-control. These studies were conducted in the USA, China, Japan, Canada, and Austria with a range of 8-320 participants
**Table 1   Characteristics of reviewed studies evaluating serum metabolites markers for colorectal CRA.**

**A. Studies reported with sensitivity and specificity values**

| Author(s)/country/ year | Instrument | Type of sample | Type of profiling | Total no. of sample | No. of participants | | | Serum metabolites marker/ Panel | Diagnostic performance between: | AUC/ ROC % | Sensitivity % | Specificity % |
|---|---|---|---|---|---|---|---|---|---|---|---|---|
| | | | | | Normal | CRA | CRC | | | | | |
| *Zhu et al. (2014)*/ USA | LC-MS | Serum | Targeted | 234 | 92 | 76 | 66 | adenosine, alanine, phosphoenolpyruvate (PEP), glyceraldehyde, glycocholate, hippuric acid, glycochenodeoxycholate, trimethylamine-N-oxide, N-acetyl glycine, hydroxyproline/aminolevulinate, dimethylglycine, linolenic acid, leucic acid, and pantothenate | CRA *vs.* CRC | 0.94 | 0.92 | 0.86 |
| *Uchiyama et al. (2017)*/Japan | CE-TOF MS | Serum | Untargeted | 175 | 60 | 59 | 56 | Benzoic acid | CRA *vs.* Normal | 0.92 | 0.88 | 0.85 |
| | | | | | | | | Benzoic acid | CRA *vs.* CRC | 0.89 | 0.89 | 0.82 |
| | | | | | | | | Lactate + Citrate | CRA *vs.* Normal | 0.8310 | 0.8125 | 0.7895 |
| | | | | | | | | | CRA *vs.* CRC | 0.7130 | 0.5750 | 0.8158 |
| *Gu et al. (2019)*/ China | 1HNMR | Serum | Untargeted | 110 | 38 | 32 | 40 | | CRA *vs.* Normal | 0.8210 | 0.7813 | 0.7368 |
| | | | | | | | | Acetate + Glycerol | CRA *vs.* CRC | 0.7720 | 0.7188 | 0.6750 |

**B. Studies without sensitivity and specificity values**

| Author(s)/country/ year | Instrument | Type of sample | Type of profiling | Total no. of sample | Normal | CRA | CRC | Serum metabolites marker/ Panel | Diagnostic performance between: | AUC/ ROC % | Sensitivity % | Specificity % |
|---|---|---|---|---|---|---|---|---|---|---|---|---|
| *Farshidfar et al. (2016)*/ Canada | GC-MS/MS | Serum | Untargeted | 605 | 254 | 31 | 320 | Cystine (4TMS) Unmatched_RI 1978 Unmatched_RI 1101 Unknown_Alkane_RI 1704 Erythritol (4TMS) Glutamic acid (3TMS) Heptadecanoic acid (1TMS) Unmatched_RI 2934 Unmatched_RI 2496 Unmatched_RI 1961 Glyceric acid (3TMS) Unknown_Alkane_RI 1431 Unknown_Alkane_RI 1448 Unmatched_RI 2379 | CRA *vs.* Normal | 0.81 | NA | NA |
| *Long et al. (2017)*/ USA | LC-MS/MS | Serum | Untargeted | 240 | 80 | 80 | 80 | Xanthine, hypoxanthine, D-mannose | CRA *vs.* CRC | NA | NA | NA |
| *Chen et al. (2017)*/USA | LC-MS/MS | Serum | Untargeted | 113 | 83 | 39 | 36 | aspartic acid, glutamine, lysine, methionine, histidine, hippuric acid, alpha-ketoglutarate, glyceraldehyde, hydroxyproline/aminolevulinate, linoleic acid, linolenic acid, 2′-deoxyuridine alanine, alanine, xanthine, and orotate | CRA *vs.* Normal | NA | NA | NA |
| | | | | | | | | d glucose-1,6-bisphosphate (G16BP), phenylalanine, xanthosine and epinephrine | CRA *vs.* CRC | NA | NA | NA |
| *Liu et al. (2023)*/ USA | Isobaric Labelling Mass spectrometry | Serum | Targeted | 43 | 20 | 23(15 high risk, 8 low risk to develop CRC) | 0 | Asparagine + Threonine | Growing CRA *vs.* Normal | 0.864 | NA | NA |
| *Tevini et al. (2022)*/ Austria | FIA and LC-MS/MS | Serum | Targeted | 235 | 93 | 76 | 66 | short chain Acylcarnitines to free carnitine ((C2 + C3)/C0) | CRA *vs.* Normal | 0.787 | NA | NA |
| | | | | | | | | PC aa/ diacyl-glycerophosphocholine C36:5 | CRA *vs.* CRC | 0.831 | NA | NA |
| *Guo et al. (2023)*/ China | UPLC-MS/MS | Serum | Untargeted | 30 | 14 | 8 | 8 | Eugenol Guanosine 5′-Diphospho-Beta-L-Fucose 3,4,5-Trimethoxybenzoiz Acid 2,6-Diaminooimelic Acid Vanillic Acid L-Ascorbate A-d-Glucose Dulcitol L-Dihydroorotic AcidE-pinephrine | CRA *vs.* Normal | NA | NA | NA |
| | | | | | | | | Hypoxanthine-9-B-D-ArabinofuranosideD-Glutamic Acid L-Rhamnose L-Fucose | CRA *vs.* CRC | NA | NA | NA |

**Table 2  List of individual/panel serum metabolites marker for CRA *vs.* normal.**

| Author(s)/country/year | Metabolites marker | Sensitivity % | Specificity % | Predictive performance between: |
|---|---|---|---|---|
| *Uchiyama et al. (2017)*/ Japan | Benzoic acid | 0.88 | 0.85 | CRA *vs.* Normal |
| *Gu et al. (2019)*/ China | Lactate + Citrate | 0.8125 | 0.7895 | CRA *vs.* Normal |
| | Acetate + Glycerol | 0.7813 | 0.7368 | CRA *vs.* Normal |

**Table 3  List of serum metabolites for CRA *vs.* normal with individual values of sensitivity and specificity extracted from reviewed studies.**

| Metabolites | Sensitivity | FPR | AUC |
|---|---|---|---|
| Acetate | 0.8 | 0.2632 | 0.7368 |
| Lactate | 0.813 | 0.2105 | 0.7895 |
| Benzoic acid | 0.88 | 0.15 | 0.85 |

Notes.
FPR, False positive rates; AUC, Area under the curve.

**Table 4  List of individual/panel serum metabolites marker for CRA *vs.* CRC.**

| Author(s)/country/year | Serum metabolites marker | Predictive performance between: | Sensitivity % | Specificity % |
|---|---|---|---|---|
| *Uchiyama et al. (2017)*/Japan | Benzoic acid | CRA *vs.* CRC | 0.89 | 0.82 |
| *Zhu et al. (2014)*/USA | Adenosine, alanine, phosphoenolpyruvate (PEP), glyceraldehyde, glycocholate, hippuric acid, glycochenodeoxycholate, trimethylamine-N-oxide, N-acetyl glycine, hydroxyproline/aminolevulinate, dimethylglycine, linolenic acid, leucic acid, and pantothenate | CRA *vs.* CRC | 0.92 | 0.86 |
| *Gu et al. (2019)*/ China | Lactate + Citrate | CRA *vs.* CRC | 0.5750 | 0.8158 |
| | Acetate + Glycerol | CRA *vs.* CRC | 0.7188 | 0.6750 |

for each group using six different analytical platforms, including LC-MS, Isoberic Labelling-MS, UHPLC-MS/MS, GC-MS, 1H-NMR, and FIA and LC-MS (Table 1). Only three studies (*Gu et al., 2019*; *Uchiyama et al., 2017*; *Zhu et al., 2014*) reported sensitivity and specificity of suggested serum biomarkers (Table 1).

## The quality of studies

The quality assessment results for the individual studies were conducted using QUADOMICS (Table S1). All studies included in this review met the inclusion criteria and scored positively which indicates that the overall quality was good.

## Serum metabolites biomarkers differentiating colorectal adenomas *vs.* normal

Seven studies (*Guo et al., 2023*; *Tevini et al., 2022*; *Liu et al., 2023*; *Chen et al., 2017*; *Uchiyama et al., 2017*; *Farshidfar et al., 2016*; *Zhu et al., 2014*) had reported a list of potential

**Table 5  List of serum metabolites for CRA *vs.* CRC with individual sensitivity and specificity values extracted from reviewed studies.**

| Metabolites | Sensitivity | FPR | AUC |
| --- | --- | --- | --- |
| Benzoic acid | 0.890 | 0.180 | 0.82 |
| Alanine | 0.864 | 0.605 | 0.3947 |
| Glyceraldehyde | 0.742 | 0.303 | 0.6974 |
| Dimethylglycine | 0.712 | 0.382 | 0.6184 |
| Hydroxyproline/Aminolevulinate | 0.667 | 0.342 | 0.6579 |
| Glycochenodeoxycholate | 0.636 | 0.342 | 0.6579 |
| N-AcetylGlycine | 0.606 | 0.303 | 0.6974 |
| Hyppuric Acid | 0.591 | 0.211 | 0.7895 |
| Glycocholate | 0.561 | 0.224 | 0.7763 |
| Linolenic Acid | 0.546 | 0.250 | 0.75 |
| Trimethylamine-N-oxide | 0.439 | 0.211 | 0.7895 |
| Adenosine | 0.394 | 0.092 | 0.9079 |
| Linoleic Acid | 0.364 | 0.079 | 0.9211 |
| Pentothenate | 0.106 | 0.066 | 0.9342 |

**Notes.**
FPR, False positive rates; AUC, Area under the curve.

individual/panel serum metabolites for CRA *vs.* Normal. However, only two of them have reported the sensitivity/specificity value (Table 2). The differential serum metabolites associated with CRA *vs.* normal identified by seven included studies in Table 1 were enriched into pathways analysis. The five most affected pathways in CRA *vs.* normal as presented in Fig. 2 were the glycoxylate and dicarboxylate metabolism; alanine, aspartate, and glutamate metabolism; aminoacyl-tRNA biosynthesis; D-glutamine and D-glutamate metabolism; and nitrogen metabolism.

The individual sensitivity and specificity values for each serum metabolite were further extracted (Table 3) and analyzed by constructing a graph of sensitivity *vs.* false positive rates (FPR) to identify their diagnostic accuracy. Glycerol and citrate were excluded from the list due to the unavailability of their individual data. Figure 3 shows the graph of sensitivity *vs.* FPR for these biomarkers where all three serum metabolites (benzoic acid, acetate, and lactate) show significant diagnostic accuracy in differentiating CRA and normal patients (AUC > 0.7).

## Serum metabolites biomarker differentiating CRA *vs.* CRC

Seven studies (*Guo et al., 2023*; *Tevini et al., 2022*; *Gu et al., 2019*; *Chen et al., 2017*; *Long et al., 2017*; *Uchiyama et al., 2017*; *Farshidfar et al., 2016*) reported individual/panel serum biomarkers associated with CRA *vs.* CRC. The differential serum metabolites reported by these studies (Table 1) were enriched into pathways analysis. Figure 4 presents the affected pathways for CRA *vs.* CRC. The five most affected pathways were pyruvate metabolism, glycolysis/gluconeogenesis, glycerolipid metabolism, citrate/TCA cycle, and alanine, aspartate, and glutamate metabolism.

Only three of the studies (*Gu et al., 2019*; *Uchiyama et al., 2017*; *Zhu et al., 2014*) have included the value of sensitivity and specificity for each reported serum biomarker CRA *vs.*

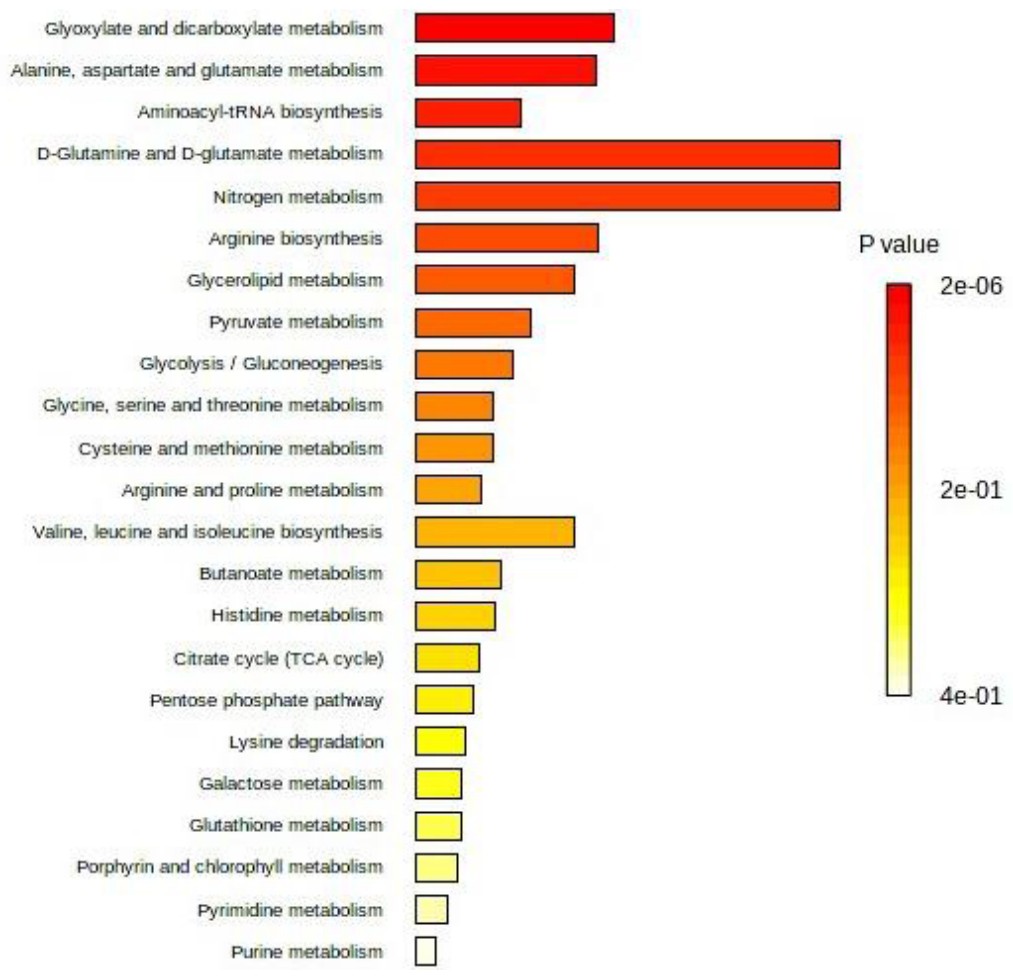

**Figure 2** **Top 25 enriched pathway analysis of metabolites marker for CRA *vs*. Normal.**

CRC (Table 4). The individual sensitivity and specificity values for each serum metabolite were further extracted (Table 5) and analyzed by constructing a graph of sensitivity *vs*. FPR to identify their diagnostic accuracy. Three of them (adenosine, pentothenate, and linoleic acid) show AUC >0.9 (Fig. 5).

## DISCUSSION

This study reviewed a total of nine publications, from January 2014 to December 2023, in five countries, using seven different analysis platforms. The review found that limited studies reported serum metabolites with sensitivity and specificity values for CRA. From the data extracted, forty-eight serum metabolites were reported to be significantly identified for CRA *vs*. Normal, and twenty-nine serum metabolites were associated with CRA *vs*. CRC. However, none of them were found to overlap between the studies. This might be caused by the diverse populations used, different analytical platforms, the diverse number of markers evaluated (single *vs*. combined panel markers), and the use of different cut-off

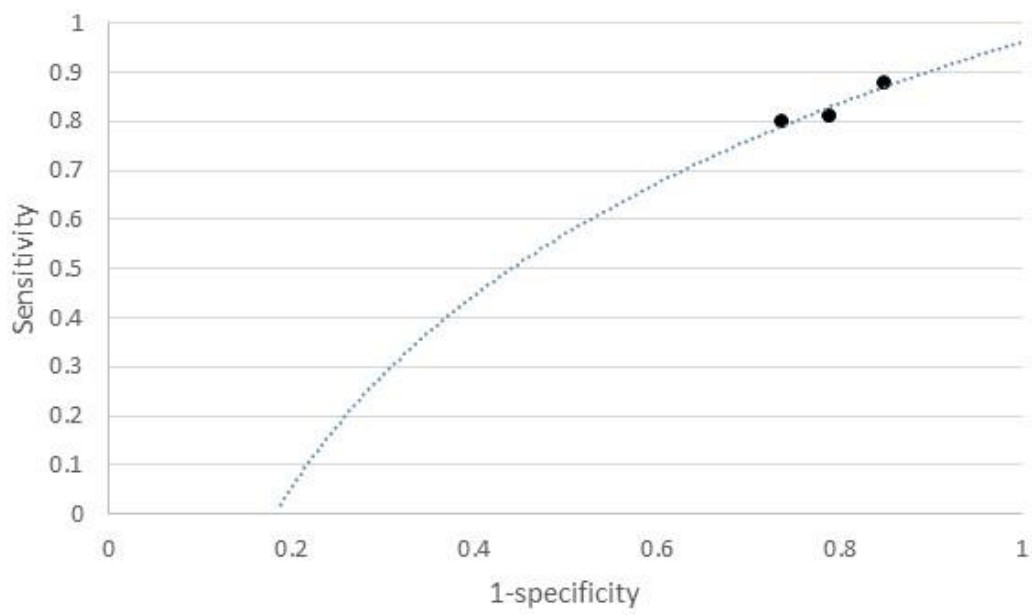

**Figure 3** ROC of serum metabolites (colorectal adenoma *vs.* normal).

points as suggested by *Kastenmüller et al. (2015)* and *Liu et al. (2014)*. Another concern can be considered by referring to the comparability of results across studies given the potential differences in serum collection, processing, and storage methods, and uncertainties in the stability of reported serum biomarkers. Information on these issues is very limited, resulting in an order-magnitude range of sensitivities and specificities reported for these markers (*Liu et al., 2014*).

The present review suggested that benzoic acid, acetate, and lactate can be used to distinguish CRA *vs.* Normal with AUC > 7.0, while adenosine, pentothenate, and linoleic acid are highly significant in differentiating CRA *vs.* CRC with AUC > 9.0. Although benzoic acid was suggested by *Uchiyama et al. (2017)* for both CRA *vs.* normal and CRA *vs.* CRC, our findings found that benzoic acid had a lower diagnostic performance of (AUC > 8.0) compared with adenosine, pentothenate, and linoleic acid with (AUC > 9.0) found by *Zhu et al. (2014)*. However, benzoic acid shows a good performance for CRA *vs.* Normal compared to lactate and acetate (AUC > 7.0) found and suggested by *Gu et al. (2019)* with AUC > 8.0. The findings also show that the AUC values for serum metabolites for CRA *vs.* normal are slightly less (range between 7.0 to 8.0 only) compared with AUC values for serum metabolites associated with CRA *vs.* CRC (up to 9.0).

According to *Martínez-Camblor, Pérez-Fernández & Díaz-Coto (2022)*, AUC values indicate a summary of the accuracy index. The higher values are usually associated with higher probabilities of having the characteristic under study. Thus, a combination of serum metabolites with a high value of AUC can be suggested to increase the diagnostic performance of serum metabolites for CRA diagnosis. For example, a combination panel of lactate, acetate, and benzoic acid can be suggested for CRA *vs.* Normal, and adenosine,

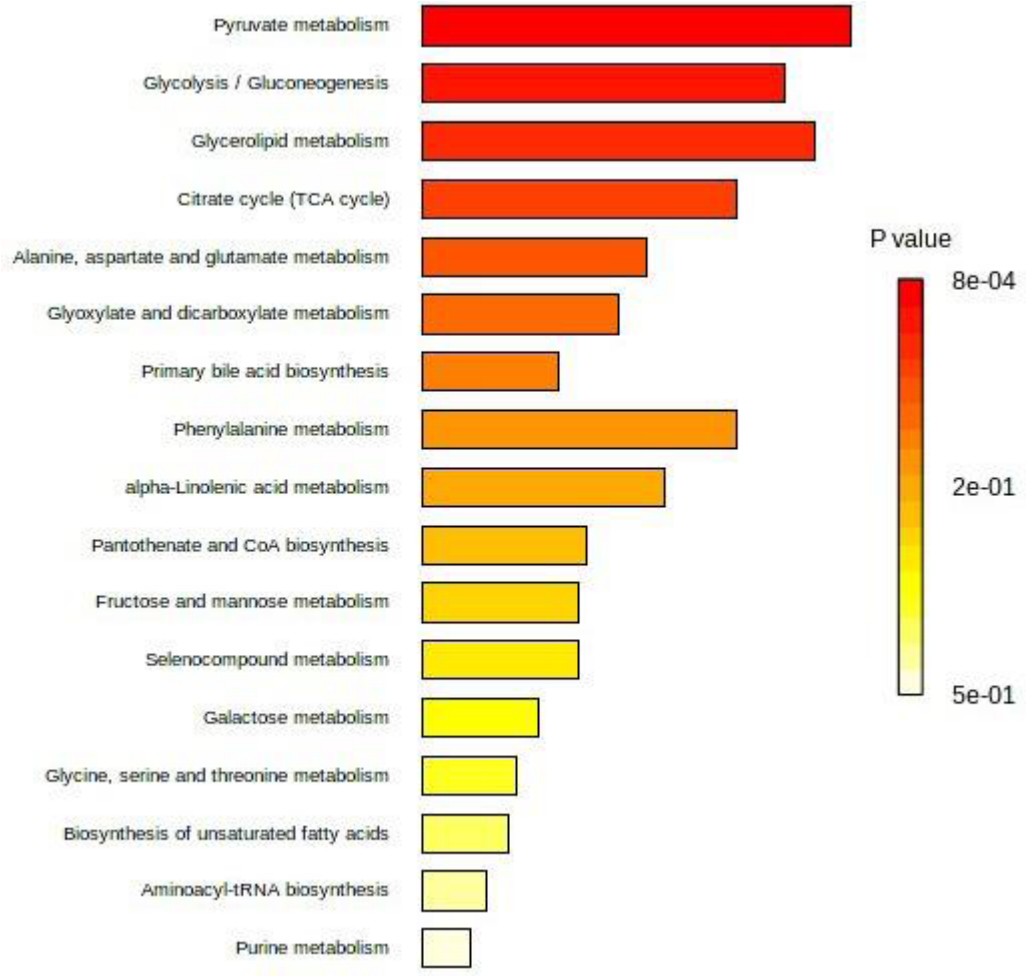

**Figure 4** Top 25 enriched pathway analysis of metabolites marker for CRA *vs.* CRC.

pantothenate, and linoleic acid in one panel for CRA *vs.* CRC. Further study is needed to investigate their accuracy in detecting CRA across populations.

For CRA *vs.* Normal, the five most affected pathways are glycoxylate and dicarboxylate metabolism; alanine, aspartate, and glutamate metabolism; aminoacyl-tRNA biosynthesis; D-glutamine and D-glutamate metabolism; and nitrogen metabolism. On the other hand, pyruvate metabolism, glycolysis/gluconeogenesis, glycerolipid metabolism, citrate/TCA cycle, and alanine, aspartate, and glutamate metabolism were found to be altered in CRA *vs.* CRC. However, the association between these altered pathways with serum metabolites reported for CRA remained unclear.

The ongoing research in this field has yielded encouraging results, with various studies highlighting the correlation between specific serum biomarkers and the presence of CRA. However, it is crucial to acknowledge that more comprehensive and large-scale clinical trials are needed to establish the accuracy, sensitivity, and specificity of these biomarkers in diverse populations. The development of serum metabolite biomarkers holds the promise
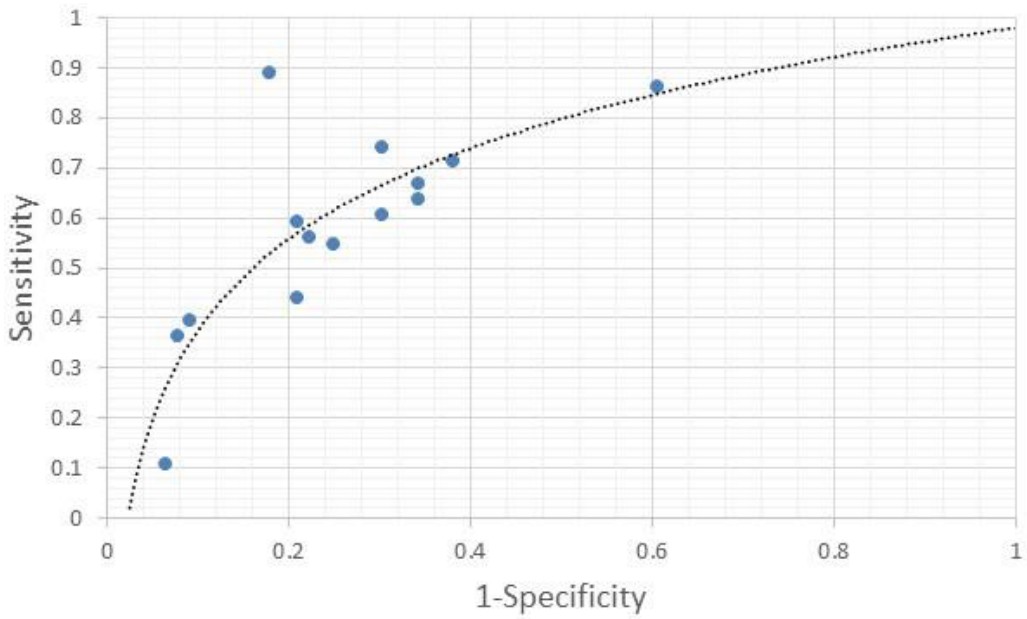

**Figure 5  ROC of serum metabolites (colorectal adenoma *vs.* CRC).**

of enhancing current screening methodologies, helping to reduce the morbidity and mortality associated with CRC by enabling early intervention and treatment. In the future, collaborative efforts among researchers, clinicians, and policymakers will be essential to advance the field of serum biomarkers for CRA. Standardization of testing protocols, validation across diverse populations, and incorporation into evidence-based guidelines will be pivotal steps to ensure the successful translation of promising biomarkers from the laboratory to clinical practice.

## Limitations

A small number of reviewed studies reporting the sensitivity and specificity of serum metabolites associated with CRA indicates a huge gap in studying serum metabolite biomarkers for CRA detection. Different analytical platforms and diverse populations in reviewed studies led to differences in signature serum metabolites identified between studies. Although a limited number of studies provided a limited data analysis of diagnostic accuracy for these serum metabolites, this review could serve as a reference for future research in selecting the best candidates of serum metabolites with reliable accuracy in detecting CRA.

## CONCLUSIONS

The exploration of serum biomarkers for CRA could give a promising return for early detection and prevention of CRC. The identification and validation of reliable serum biomarkers have the potential to revolutionize current screening practices for CRA and CRC, offering a non-invasive and cost-effective means to detect CRC at an earlier, more

treatable stage. While the journey towards establishing serum biomarkers for CRA and CRC as routine clinical tools is still ongoing, it is important to ensure the accuracy of each serum metabolite reported in detecting the disease. Continued research, validation, and implementation efforts are crucial to enhance the full potential of serum biomarkers in transforming the landscape of CRC screening and improving patient outcomes.

## ACKNOWLEDGEMENTS

We would like to express our gratitude to the Department of Public Health Medicine, Universiti Kebangsaan Malaysia for their technical guidance in conducting this study.

### Funding

The work was supported by the Fundamental Research Grant Scheme (FRGS), grant number FRGS/1/2021/SKK05/UKM/02/1, funded by the Ministry of Higher Education (MOHE), Malaysia. The funders had no role in study design, data collection and analysis, decision to publish, or preparation of the manuscript.

### Grant Disclosures

The following grant information was disclosed by the authors:
the Fundamental Research Grant Scheme (FRGS): FRGS/1/2021/SKK05/UKM/02/1.
The Ministry of Higher Education (MOHE), Malaysia.

### Competing Interests

The authors declare there are no competing interests.

### Author Contributions

- Maryam Fatimah Abu Bakar conceived and designed the experiments, performed the experiments, analyzed the data, prepared figures and/or tables, authored or reviewed drafts of the article, and approved the final draft.
- Siok Fong Chin performed the experiments, analyzed the data, authored or reviewed drafts of the article, and approved the final draft.
- Suzana Makpol conceived and designed the experiments, performed the experiments, analyzed the data, prepared figures and/or tables, authored or reviewed drafts of the article, and approved the final draft.
- Jen Kit Tan performed the experiments, analyzed the data, authored or reviewed drafts of the article, and approved the final draft.
- Azmawati Mohammed Nawi conceived and designed the experiments, performed the experiments, analyzed the data, prepared figures and/or tables, authored or reviewed drafts of the article, and approved the final draft.

### Data Availability

This is a systematic review/meta-analysis.

## Supplemental Information

Supplemental information for this article can be found online at http://dx.doi.org/10.7717/peerj.18043#supplemental-information.

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
