# Peer review of "Diagnostic performance of serum metabolites biomarker associated with colorectal adenoma: a systematic review"

_PeerJ, doi:10.7717/peerj.18043_

## Round 0.1 · accepted · Accept

# The manuscript was reviewed by two referees. Based on their comments and my own reading of the manuscript, I believe that it is publishable.

Reviewer 1 ·

Basic reporting

Background:
The background section provides sufficient context on CRA and its significance in preventing CRC, with a well-justified focus on non-invasive diagnostic tools.

Methodology:
The methodology is robust, featuring a comprehensive database search, adherence to PRISMA guidelines, use of QUADOMICS for quality assessment, and MetaboAnalyst 5.0 for data analysis, enhancing the study's credibility.
Key Findings:
The identification of specific biomarkers with AUC values between 0.7 and 1 is promising, and the detailed analysis of affected pathways offers valuable insights into the biological mechanisms involved.
Conclusion:
The conclusion effectively summarizes the findings, highlighting the need for further validation, and aligns well with the study's objective.
Manuscript Structure:
The structure is standard and well-organized, making the manuscript easy to follow. Figures and tables are relevant, high-quality, and well-labeled, enhancing data presentation.
Language and Presentation:
The manuscript is written in clear and professional English. The introduction and background sections provide sufficient context, and the literature is well-referenced. Figures are high-quality, and raw data is supplied as required by the journal.
Review Notes:
The review notes are constructive, emphasizing clarity and precision in language, validation of biomarkers in diverse cohorts, and standardization of methodologies across studies for better comparability and reliability of results.
Overall Assessment:
The manuscript is well-prepared, with a clear objective, robust methodology, and promising findings. The authors should focus on further validating their findings and ensuring methodological consistency across future studies.

Experimental design

The manuscript presents a robust experimental design, with a comprehensive search of PubMed, Scopus, and Web of Science from January 2014 to December 2023, adhering to PRISMA guidelines. The QUADOMICS tool for quality assessment enhances credibility, though more details on its application and challenges are needed. The extraction and analysis of 77 biomarkers using MetaboAnalyst 5.0 is comprehensive, but the manuscript should elaborate on the criteria for selecting relevant biomarkers and pathways. ROC curves for diagnostic accuracy are effective, but threshold values and rationales should be included. Acknowledging limitations such as variability in study designs, populations, and methodologies will provide a balanced view. Further validation of identified biomarkers in diverse populations with standardized methods is crucial. Emphasizing reproducibility through detailed protocols and criteria is necessary. Ethical considerations should be mentioned, especially for studies involving human subjects. Overall, the experimental design is strong, but enhancements in transparency, detailed descriptions, acknowledgment of limitations, and emphasis on reproducibility will strengthen the study's reliability and applicability.

Validity of the findings

The manuscript demonstrates a thorough literature search across PubMed, Scopus, and Web of Science using PRISMA guidelines, ensuring a broad, systematic approach that minimizes selection bias and enhances the validity of the findings. The use of the QUADOMICS tool for quality assessment ensures high-quality evidence by filtering out studies with methodological weaknesses. Data analysis with MetaboAnalyst 5.0 supports the validity of the findings through robust statistical capabilities for identifying altered pathways and potential biomarkers. Constructing ROC curves to evaluate diagnostic accuracy based on sensitivity and specificity provides a reliable measure, further supporting the findings' validity. The identification of six serum metabolite biomarkers with AUC values between 0.7 and 1 is significant but requires further validation in larger, diverse studies, as current findings are promising yet preliminary. The pathway analysis reveals affected biological pathways in CRA vs. normal and CRA vs. CRC comparisons, adding biological plausibility, though potential methodological variations across studies could impact results. The manuscript should acknowledge potential limitations, such as variability in study design, population characteristics, and methodologies, which can introduce bias and affect generalizability. Emphasizing the need for further validation of identified biomarkers in larger, diverse populations with standardized methodologies is crucial for confirming findings and enhancing validity. Comparing findings with existing literature on CRA biomarkers can help validate results, with consistency strengthening validity and discrepancies warranting discussion for potential reasons and implications. Overall, the findings are valid within the scope of included studies and methodologies but require further validation in extensive, diverse cohorts. Transparent acknowledgment of limitations and biases, alongside a clear path for future research, will strengthen the findings' overall validity and applicability.

Reviewer 2 ·

Basic reporting

No comment.

Experimental design

No comment.

Validity of the findings

No comment.

Additional comments

In this article, the authors conducted a systematic search to identify the diagnostic accuracy of serum metabolites associated with CRA. The manuscript is straightforward, well written, and concise and has clear results within the scope of a review article. Definitely deserves to be published and is a valuable contribution to the “PeerJ” journal. However, the following comments need to be addressed before publication, as per recommended.

[1] “Introduction”, Lines 50-51:
“CRC ranked as the second cancer with high mortality and incidence rates reported around the world (IARC., 2020).”.
From the epidemiological point of view, the authors should mention that very recently, new favorable subsets of cancers of undefined origin (CUP) seem to emerge, including colorectal CUP. This new clinical entity is treated as CRC, and contributes to the current increased incidence of CRC.
Recommended reference: Rassy E, et al. New rising entities in cancer of unknown primary: Is there a real therapeutic benefit? Crit Rev Oncol Hematol. 2020 Mar;147:102882.

[2] “Introduction”, Lines 58-60:
“Several efforts to discover serum metabolite biomarkers also have been made and a few potential serum metabolite biomarkers for CRC have been reported.”.
This should be discussed in more detail, as biomarker testing is recommended as a part of the standard investigation in CRC. Among several major genetic mutations in CRC, RAS mutation is correlated with the oncological aggressiveness and the pathologic response to chemotherapy. There is also growing evidence that inflammation drives development of the disease. As a result, many studies have investigated the predictive and prognostic role of various blood based inflammatory markers, including neutrophil–lymphocyte ratio (NLR), lymphocyte–monocyte ratio (LMR), and platelet–lymphocyte ratio (PLR). Finally, miRNAs have roles as tumor suppressor genes and oncogenes, and their diagnostic, prognostic, and predictive implications are now being explored.
Recommended reference: Boussios S, et al. The Developing Story of Predictive Biomarkers in Colorectal Cancer. J Pers Med. 2019(1);9:12.

[3] “Discussion”, Lines 228-230:
“The development of serum metabolite biomarkers holds the promise of enhancing current screening methodologies, helping to reduce the morbidity and mortality associated with CRC by enabling early intervention and treatment.”.
At that point, the authors should mention that immune cell PD-L1 expression is significantly higher in mismatch repair (MMR)-deficient (microsatellite instability high, MSI-H) CRC as compared to MMR-proficient (MSI low) tumors, with no differences among the different MSI-H molecular subtypes. The recommended screening for defective, DNA mismatch repair includes immunohistochemistry (IHC) and/or MSI test. However, there are challenges in distilling the biological and technical heterogeneity of MSI testing down to usable data. It has been reported in the literature that IHC testing of the mismatch repair machinery may give different results for a given germline mutation and has been suggested that this may be due to somatic mutations.
Recommended reference: Adeleke S, et al. Microsatellite instability testing in colorectal patients with Lynch syndrome: lessons learned from a case report and how to avoid such pitfalls. Per Med. 2022;19(4):277-286.